

# Chemical characterization of atmospheric aerosols at a high-altitude mountain site: a study of source apportionment

Elena Barbaro[1,2], Matteo Feltracco[2], Fabrizio De Blasi[1,2], Clara Turetta[1,2], Marta Radaelli[2], Warren Cairns[1,2], Giulio Cozzi[1,2], Giovanna Mazzi[2], Marco Casula[1], Jacopo Gabrieli[1,2], Carlo Barbante[1,2], Andrea Gambaro[2,1]

[1] CNR-Institute of Polar Sciences (ISP-CNR), 155 Via Torino, 30170 Mestre, Italy;
[2] Department of Environmental Sciences, Informatics and Statistics, Ca' Foscari University of Venice, Venice, Italy

\* *Correspondence to*: Matteo Feltracco (matteo.feltracco@unive.it)

**Abstract.**

The study of aerosol in high mountain regions is essential because particulate matter can play a role in altering the energy balance of high mountain regions, and aerosols can accelerate glacier melting in high mountain areas by darkening the ice surface, reducing its reflectivity (albedo). Studying aerosols in high mountain areas provides insights into long-range transport of pollutants, atmospheric dynamics, and climate change impacts. These regions can serve as valuable observatories for studying atmospheric processes.

The main aim of this paper is to define the main sources of aerosol over an entire year of sampling at the Col Margherita Atmospheric Observatory (MRG 46° 22' 0.059'' N, 11° 47' 30.911'' E, 2543 m a.s.l.), a high-altitude background site in the Eastern Italian Alps. Here, we discuss the potential origins of more than one hundred chemical markers (major ions, water soluble organic compounds, trace elements, rare earth elements) using different approaches. Some diagnostic ratios were applied, but source apportionment using Positive Matrix Factorization was used to define the main inputs of $PM_{10}$ collected at this high-altitude site. Moreover, a characterization of the air masses helped us to confirm the aerosol sources.

Keywords: aerosol composition, source apportionment, air masses, high mountain





## 1. Introduction

Atmospheric aerosol have strong impacts on the climate (Ren-Jian et al., 2012) because these particles can absorb and diffuse solar radiation, impacting the radiation budget. Their chemical composition can influence their hygroscopic proprieties and hence their ability to act as cloud condensation nuclei (CCN) (Hitzenberger et al., 1999). Aerosol particles can also impact the environment, influencing air pollution. The chemical composition of these particles can have specific impacts on human health, producing respiratory disorders, strokes, and pulmonary and cardiovascular diseases (Ren-Jian et al., 2012).

The aerosol depends on local and regional emission sources, but it can also derive by atmospheric long-range transport. These particles can be directly emitted on the atmosphere by natural sources (i.e., sea salts or crustal particles) or by anthropogenic activity such as industry or vehicular traffic. Secondary aerosol is another important component of particulate matter because several gas-to-particles or particle-phase reactions can occur, and several aging or oxidative processes can modify their chemical proprieties.

High altitude mountain stations are considered the best sites to investigate the background levels of trace gases and aerosols, due to their distance from anthropogenic emission sources. The data collected in these stations can be used to study long-range transport of dust or anthropogenic and biomass burning pollutants from emission regions. Okamoto and Tanimoto (2016) summarized the results from past and ongoing field measurements of atmospheric constituents at high-altitude stations across the globe with a particular focus on trace gases such as ozone. The authors identified 31 stations at high altitude above 1500 m a.s.l. that provide reliable observational data, online available or published in previous studies. Many stations are located in northern mid-latitudes, particularly in central Europe and western North America. Compared to the Northern Hemisphere, significantly, fewer stations are in the Southern Hemisphere, and no high-altitude stations are present in the Oceania region. One of the final goals of this publication is to propose Col Margherita Observatory as high-altitude background observatory of atmospheric aerosols in the Eastern Italian Alps.

The main aim of this paper is to characterize different source of the atmospheric aerosol collected at the Col Margherita Atmospheric Observatory (MRG 46° 22' 0.059'' N, 11° 47' 30.911'' E, 2543 m a.s.l.), a high-altitude background site in the Eastern Italian Alps. The chemical composition was used to define specific sources or processes, by combining chemometric singular specific diagnostic ratios. More than one hundred chemical markers determined: inorganic ions (Cl-, Br-, NO3-, SO42-, K+, Mg2+, Na+, NH4+, Ca2+), twelve organic acids (methanesulfonic acid and C2–C7 carboxylic acids), seven monosaccharides (arabinose, fructose, galactose, glucose, mannose, ribose, xylose), eight alcohol-sugars (arabitol, erythritol, mannitol, ribitol, sorbitol, xylitol, maltitol), three anhydrosugars (levoglucosan, mannosan and galactosan), sucrose, eleven phenolic compounds, forty free L- and D-amino acids and two photo-oxidation products of α-pinene (cis-pinonic and pinic acid), twenty seven trace elements (TE) and fifteen rare earth elements (REE). The novelty of this paper is in the combination of a consistent number of chemical species to characterize the aerosol sources, by considering different types of markers for biogenic, anthropogenic, crustal, and other type of the sources at a high mountain site. Positive matrix factorization (PMF) was performed to 1) identify the sources and to define 2) which chemical species were characteristic for each source, because the sources of some compounds are still not well known.



## 2. Experimental section

### 2.1 Sampling and meteorological conditions

A year of sampling atmospheric aerosol was performed from the 10th of August 2021 to 22nd of July 2022 at Col Margherita Atmospheric Observatory (MRG, (46° 22' 0.059'' N, 11° 47' 30.911'' E, 2543 m a.s.l.). Thanks to its location, MRG is an ideal site for the investigation of the atmospheric circulation on a regional scale.

Samples were collected using a low volume aerosol sampler (Skypost, Tecora) equipped with a sequential sampling module (average flow rate of 38.3 L min$^{-1}$) and a sequential sampling module for automatic filter changing. PM$_{10}$ samples was collected on quartz fiber filters (QFFs, Filtros Anoia, Barcelona), previously decontaminated by a 4 h pre-combustion at 400 °C in a muffle furnace. The sampling resolution was set to 96 h as a good balance between quantifying the target species at trace levels and sampling resolution. During the sampling periods, field blanks were taken at the beginning, during and at the end by loading a filter into the filter holder of the sampler for 5 min with the vacuum pump turned off.

Meteorological sensors are installed on a compact aluminium tower at 3 m above ground level (ATW3, Campbell Scientific), close to the observatory. The measured parameters were: air temperature (T) and relative humidity (RH), accuracy ± 0.1 °C and ± 2%, respectively (CS215, Campbell Scientific); atmospheric pressure (P), accuracy ± 0.3 hPa at +20 °C (PTB110 barometer, Vaisala); snow depth (SnD), accuracy ± 1 cm (Sonic Ranging Sensor SR50A, Campbell Scientific). The sampling frequency was every 5 min, but hourly averages were used for data analysis.

During the considered period, we have registered over the 89% of the hourly data. We used the procedure described by Carturan et al. ( 2019) to fill the gaps in the time series. The reference weather station for gap filling is the one located at Passo Valles, about 3.3 km away and 500 m lower. Fig. S1 shows the average and extreme values for each meteorological parameter that we have investigated.

### 2.2 Mass concentration and chemicals analysis

Blank and sample filters were weighed three times (%RSD 5–10%) before and after sampling using a KERN ALT 220-4NM balance (readability 0.1 mg, repeatability SD ± 0.1 mg). The balance and the filters were kept in a temperature (25°) and humidity-controlled (RH 50%) nitrogen glove box.

Half of the filter was broken into small pieces with a ceramic cutter and placed in a 15 mL vial (previously cleaned with ultrapure water in an ultrasonic bath for 30 min). The sample was spiked with internal standards and extracted with 5 mL of ultrapure water in an ultrasonic bath for 30 min at 10 °C to avoid volatilization of the analytes. The extract solution was filtered through a 0.45 μm PTFE filter to remove particles and quartz fiber traces before instrumental analysis. This extract was analyzed to determine inorganic ions and organic acids (methanesulfonic acid and C2–C7 carboxylic acids) (Feltracco et al., 2021b), monosaccharides, alcohol-sugars, levoglucosan and its isomers, sucrose (Barbaro et al., 2015b), free amino acids (Barbaro et al., 2015a), phenolic compounds (Zangrando et al., 2013), and photo-oxidation products of α-pinene (Feltracco et al., 2018).

The other half was microwave-digested (Ethos1-Milestone), with 6mL HNO3, 3 mL H2O2, and 1 mL HF (Romil®UPA) in PFA high pressure digestion vessels. The digestion temperature program consisted of a ramp from room temperature to 190 °C in 25min, after which this value was maintained for 15 min. One or two blank samples per digestion batch containing the same amount of the individual acids were during sample digestion. The analysis of 27 TE (Li, Be, Mg, K, Ca, V, Cr, Mn, Fe, Co, Ni, Cu, Zn, Ga, As, Y, Rb, Sr, Ag, Cd, Cs, Ba, Tl, Pb, Bi, U and Th) and 14 REE (La, Ce, Pr, Nd, Sm, Eu, Gd, Tb, Dy, Ho, Er, Tm, Yb and Lu) was performed by ICP-SFMS (Element XR Thermo Scientific, Bremen) following the method described by Turetta et al. (2021)



**2.3 Source apportionment using Positive Matrix Factorization (PMF)**

To gain a comprehensive understanding of the complete dataset and acquire insights from the high number of species investigated in this paper, we employed the Positive Matrix Factorization (PMF) method developed by Paatero and Tapper (1994). The primary objective of utilizing the PMF technique was to determine the count of underlying factors or sources,
their chemical compositions, and the corresponding mass contributions. These factors/sources played a role in shaping the observed PM concentrations. For this study, we utilized the EPA PMF 5.0 code, which applies the Multilinear Engine (ME-2) for its implementation.

In this study, we examined a dataset comprising of the concentrations of identified chemical species (n=37), $PM_{10}$ mass concentration and 85 aerosol samples. These ranges were treated as independent samples of the same species (n x m = 37
× 85, where n represents each chemical species and m represents the number of cases or samples). The uncertainties of each sample concentration were defined using the relative standard deviation for each variable during the validation process and an additional 10% modelling uncertainty was applied during the elaboration.

To determine the uncertainties in the PMF outcomes, the bootstrap method was employed. The classification of the chemical species was carried out using the Signal-to-Noise criteria (S/N) as outlined by Paatero and Hopke (2003).
Moreover, we also utilized the percentage of data surpassing the detection limit as a supplementary criterion, following the approach introduced by Amato et al. (2016). This combined analysis underscored the robust nature of all the identified chemical species.

**2.4 Air masses back-trajectories**

To understand the long-range transport pattern of air masses recorded at the sampling site, we conducted a new back
trajectory analysis going back 7-day every 6 h. The elevation of the site was considered the elevation of site (2543 m a.s.l) plus an extra 1000 m to avoid problems from the surrounding orography. The NOAA HYSPLIT trajectory model from the NOAA ARL was applied, using the GDAS one-degree meteorological database in the Hybrid Single-Particle Lagrangian Integrated Trajectory (HYSPLIT) model (Draxler, 1998). Based on the backward particle release simulation, the cluster aggregation was displayed for each 4-day sampling period, considering the total spatial variance.

**3.    Results and discussion**

**3.1 Seasonal trend of $PM_{10}$ in the alpine site of Col Margherita**

Mass concentrations of $PM_{10}$ for a full year were evaluated at the high-altitude Alpine site of Col Margherita (Fig. 1). The concentrations ranged between 0.5 and 180 µg m$^{-3}$, with a median value of 6 µg m$^{-3}$. A huge concentration spike of 180 µg m$^{-3}$ (sample between 15[th] and 19[th] March) was found due to the intrusion of an air mass containing Saharan dust. The
5-days back trajectories (Fig. S2) show the air masses came from the Sahara Desert, crossed Spain, and France at low altitude, before arriving at the MRG sampling site.
Higher mean concentrations were found during summer 2021 (7±4 µg m$^{-3}$) and spring (9±5 µg m$^{-3}$) and summer 2022 (12±6 µg m$^{-3}$) than those during autumn 2021 and winter 2021-2022, where the mean concentrations were 3±3 µg m$^{-3}$ and 5±3 µg m$^{-3}$, respectively. This seasonal trend with higher values during the spring and summer seasons was also
found in more than 20 years of measurements at the Jungfraujoch Research Station (Bukowiecki et al., 2016) in the



Western Swiss Alps (3580 m asl), as well as during 1998- 2011 at the Mt. Cimone observatory (Tositti et al., 2013) in the Italian northern Apennines (2165 m asl).

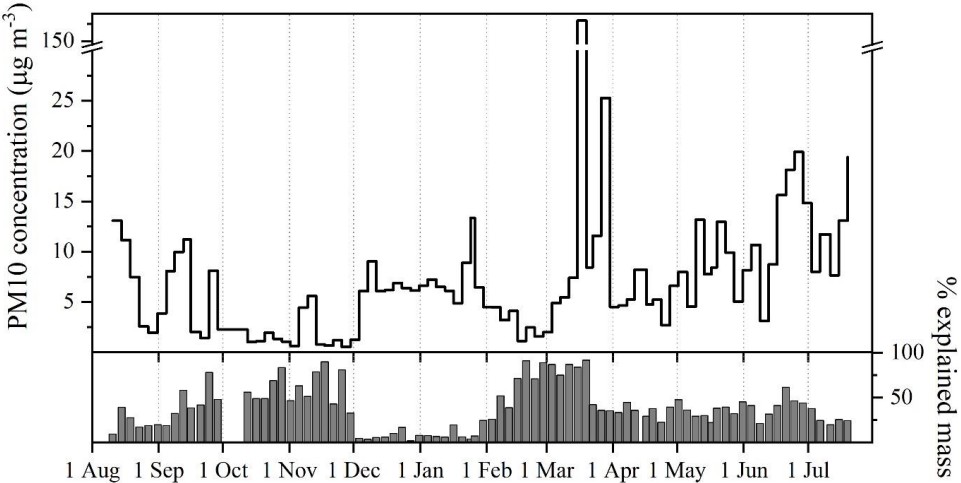


**Figure 1. A) Annual trend of PM₁₀ mass concentrations at MRG from August 2021 to July 2022. B) Percentages of explained mass obtaining by the comparison of the sum of analytes determined in filters with the mass concentration.**

This well-known seasonal trend was confirmed in our year of measurements at MRG. It is due to a combination of several

atmospheric processes: thermal convection and a mountain/valley breeze regime, and upward motion due to mixed layer expansion. The effect of valley breezes was evaluated in our recent publication (Feltracco et al., 2022), demonstrating that wind typically arises in the late morning of warm season days (July and August), and then channels upward to the MRG site, interacting with the local up-valley flow. This effect is much less present during the cold season. However, at Jungfraujoch (Bukowiecki et al., 2016), vertical transport of the PBL was found to be mostly responsible for this

seasonality, thanks to observations of in-situ aerosol parameters and the aerosol optical depth (Ingold et al., 2001). The impact of the PBL was evaluated at MRG using ERA5 (Vardè et al., 2022) during different seasons in 2018-2019, finding a diurnal variability with a lowest PBL height in winter.

Figure 1B underlines the percentage of PM₁₀ mass concentration explained through the analysis of all species considered in this study. During the spring, around 50% of the PM₁₀ is explained while the PM₁₀ is well characterized in February-

March with values close to 100%. In December and January, the percentages of explained mass were found to be between 4% and 25% . An evaluation a hypothesis on the composition of the possible missing contributions will be made further in this manuscript, during the discussion of each class of analyzed species.

### 3.2 Chemical composition

This paper shows for the first time an extended chemical characterization of atmospheric aerosol at a high-altitude

sampling site. For the whole year of samples, 100 species showed concentrations always above the method detection





limits, these were: 5 cations (Na+, NH4+, K+, Mg2+, Ca2+); 4 inorganic anions (Cl-, NO3-, SO42-, Br-); methansulphonic acid; 2 photodegradation products of α-pinene (pinic and pinonic acids); 11 carboxylic acids (formic, acetic, glycolic, oxalic, malonic, succinic, malic, maleic, fumaric, glutaric, adipic acids); 18 L- and D- free amino acids, 9 phenolic compounds (vanillic and isovanillic acids, vanillin, acetovanillone, syringic acids, syrigaldehyde,

acetosyringone, conypheryl aldehyde, ferulic acid); 3 anhydrosugars (levoglucosan, mannosan and galactosan); 7 alcohol sugars (arabitol, mannitol, erythrytol, sorbitol, xylitol, ribitol, maltitol); 7 monosaccharides (glucose, fructose, mannose, arabinose, galactose, ribose, xylose); sucrose; 26 TE; 15 REE.

Sodium, calcium and magnesium are three elements determined with two different analytical methods: the data obtained by ion chromatography coupled with a conductivity detector reflected the concentration of the ionic species of the element

in the water-soluble fraction, while the concentrations of the same elements determined on samples extracted in acid media and analyzed using ICP-SFMS described the total concentration of these elements including a portion of the insoluble fraction (Fig. S3). This explains why the concentrations obtained by ICP-SFMS are obviously higher than the ionic fractions. The crustal particles are dissolved with acids, and any fine particles present below an aerodynamic diameter of 8µm act as a solution aerosol in the spray chamber and plasma and are atomized in the plasma of the ICP-

SFMS(Goodall et al., 1993; Tong and Guo, 2019).

A comparison of the chemical compositions of each season is reported in Fig. 2, by considering the total Na, Ca and Mg, as analyzed with ICP-SFMS, and excluding the samples contaminated by the Saharan Dust event (SDE). Sulfate and nitrate are the most abundant species in the alpine aerosol that was collected each season, other crustal elements, such as Al, Ca, Na, Mg and Fe, were also abundant and represented together with the anions mentioned above the majority of the

chemical species in the aerosol collected. These abundances are consistent with the chemical composition of aerosol described at Jungfraujoch station (Cozic et al., 2008), although for a much smaller number of species (mainly major ions, organic material and black carbon). As shown in Fig. 2, some trace elements such as Fe and Al, which describe the crustal sources together Ca, are especially abundant components of these aerosols, during the seasons without snow cover.

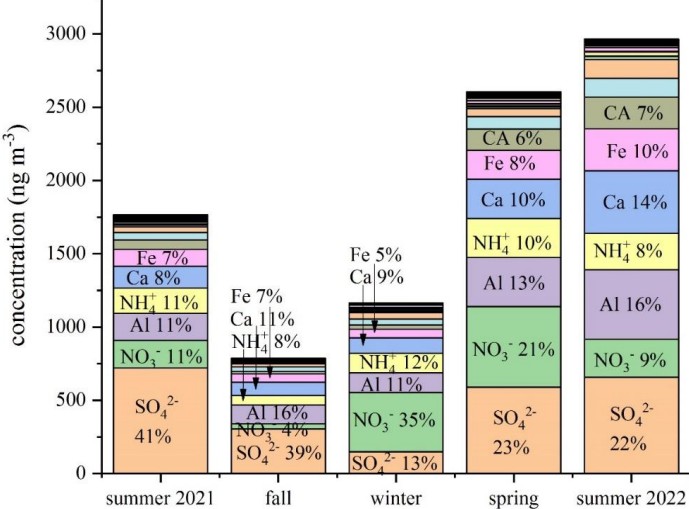


**Figure 2. Comparison of chemical composition of the aerosol collected from August 2021 to July 2022 at Col Margherita observatory in the different seasons.**



### 3.2.1 Ionic composition and ionic balance

Ionic species represent the major percentage of the total chemical composition (Fig. 2). The highest concentrations are
found in spring and summer, while during the snow season low ion concentrations are detected. Sulfate, ammonium and
nitrate are the most abundant species in the aerosol samples collected in the Eastern Alps at MRG, as also found at the
Jungfraujoch (Western Aps) (Bukowiecki et al., 2016). As demonstrated by Cozic et al. (2008), sulfate and nitrate are
neutralized by ammonia. Figure 3 reports the equivalent concentrations of each ion over the entire year and the equivalent
sum of the analysed cations/anions ratio. From November to February, the total anion concentration exceeded the total
cation concentration (neutral situation with ratio=1), meaning the aerosol was probably acidic due a significant
(unmeasured) concentration of hydrogen cations presents in the aqueous solution associated with the anions in the
aerosols. Aerosol acidity depends on a strong acid content, mainly from sulfuric and nitric acids, whose precursors occur
both in gaseous and aqueous phases (Squizzato et al., 2013). On the contrary, during the spring-summer period an anion
deficit in our measurements is recorded, plausibly due to the presence of carbonate (an unmeasured anion), emitted in the
atmosphere by the erosion processes of the Dolomite rocks (ideally CaMg(CO3)2).

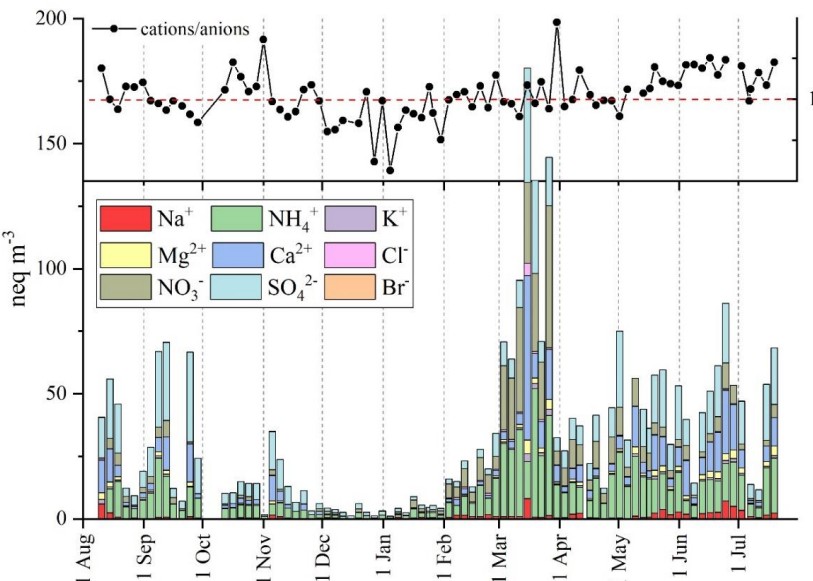

**Figure 3. Ion concentrations (expressed in neq m⁻³) determined in the annual aerosol sampling from August 2021 to the end of July 2022 at Col Margherita Observatory.**

Sulfate is one of the most concentrated ionic species in MRG aerosol for the entire sampling campaign with concentrations
between 31 ng m⁻³ and 2 µg m⁻³. Although sulfate can be emitted to the atmosphere by sea salt or produced by the
oxidation of dimethyl sulfide (DMS) released from marine algae (Gondwe et al., 2003), the anthropogenic input from the
oxidation of SO2 has already been confirmed as the main source of sulfate in atmospheric aerosol collected at MRG
(Barbaro et al., 2020). The mean anthropogenic contribution (ex SO42-), obtained by excluding sea-salt sulfate and
mineral dust sulfate contributions (Schwikowski et al., 1999), resulted as 92±6%, demonstrating that the other sources



are minor but present. For example, biogenic sources were confirmed by the significant correlation (R person 0.61, p value <0.05) between non-sea-salt-$SO_4^{2-}$ and methansulfanate (MSA) was found, confirming by the seasonal trends (Fig. S4) and higher concentrations during the summer period. The reasons for this are ascribed to the presence of biogenic inputs (Els et al., 2020) and the valley breeze and PBL impacts on the $PM_{10}$ concentrations (as previously discussed).

Nitrate is another very abundant ion in the MRG aerosol samples. The nitrate to sulfate ratio (r) in equivalents is comparable (Fig. 4) with other data in different environments. At MRG, the r average value is 0.3±0.2, except for the period between 16th January to 26th March when the mean value was 2±1 (Fig. 4). This ratio is strongly linked to distance from emission sources. Henning et al. (2003) compared the r value at Jungfraujoch (r= 0.2) with r=0.7 at an urban site in Zurich (Hueglin et al., 2005), other r values reported are those of r=1.1 in Milan (Putaud et al., 2002) and r= 1.6 for a

coastal polluted area of England (Yeatman et al., 2001). In general, remote areas or high mountain sites (Huebert et al., 1998; Preunkert et al., 2002; Shrestha et al., 1997) showed values similar to those found at MRG, except for during January-March. These low values are likely due to a faster oxidation of $NO_x$ than $SO_2$, leading to r values that are higher close to the sources. The higher values found at MRG mainly in the winter period could plausibly be due to local emissions of $NO_x$ from increased road traffic linked to tourism, and domestic heating.

Nitrate is neutralized by ammonium and these particles of ammonium nitrate derive from the conversion of $NO_x$ to nitric acid through photochemical processes and nighttime heterogeneous chemistry (Stockwell et al., 2000). The presence of nitric acid and high ammonia concentrations produces ammonium nitrate, especially in conditions of low temperature and high humidity. With low $NH_3$ concentration, $NH_4^+$ is neutralized by sulfate (Pathak et al., 2009).

The molar ratio between ammonium and sulfate is commonly used to define ammonium nitrate formation under different

environmental and chemical conditions. Several studies (Arsene et al., 2011; Huang et al., 2011; Pathak et al., 2009) define a threshold value of 1.5 for the ammonium-to-sulfate ratio needed to stabilize nitrate with ammonium, while, with $[NH_4^+]/[SO_4^{2-}] < 1.5$, nitrate neutralization can depend on gas phase reactions with nitric acid, saline or crustal particles or by heterogeneous hydrolysis of $N_2O_5$. In the Po Valley, Squizzato et al. (2013) suggested another ratio by plotting the nitrate-to-sulfate molar ratio as a function of ammonium to-sulfate molar ratio. They found that each mole of sulfate

removes 2 moles of ammonium as solid or aqueous $(NH_4)_2SO_4$. Considering the dataset from an entire year of sampling at MRG, we found a slope of 0.71±0.04 with Pearson's r of 0.89 (Fig. 4) for a plot of the molar ratios of ammonium/sulfate against the molar ratios of nitrate/sulfate, suggesting that at this remote site, one mole of ammonium neutralizes one mole of sulfate. The information obtained with the r value are confirmed even for the period between January and March when an extra input of ammonium nitrate was demonstrated.





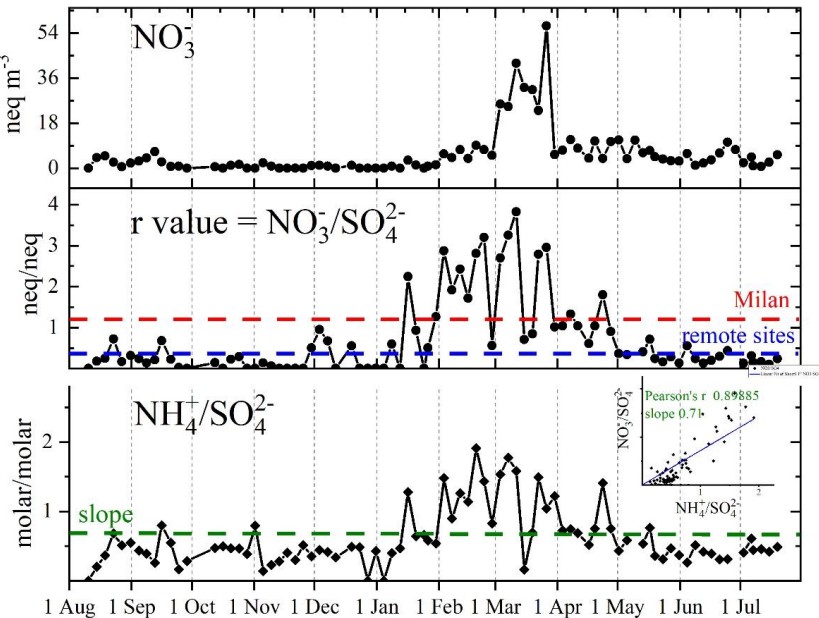

**Figure 4. A) Nitrate concentrations in neq m$^{-3}$ during the entire sampling period 2021-2022. B) trend of ratio in neq of nitrate and sulfate and C) ratio in molar between ammonium and sulfate.**

### 3.2.2 Secondary aerosol: carboxylic acids and the photo-oxidation products of α-pinene

The annual trend of twelve organic acids (C2-acetic, C2-glycolic, C2-oxalic, C3-malonic, C4-succinic, cis-C4-maleic, trans-C4-fumaric, hC4-malic, C5-glutaric, C6-adipic, pinonic and pinic acids) showed that the concentrations in spring (150±107 ng m$^{-3}$) and in summer (70±61 ng m$^{-3}$ for 2021 and 222±124 ng m$^{-3}$ for 2022) were higher than fall (22±19 ng m$^{-3}$) and winter (32±41 ng m$^{-3}$). Many organic acids are not found in the cold period while only pinonic and pinic acids, the photooxidation products of α-pinene, are found in these seasons (Fig. S5). The concentrations of summer and spring period are consistent with those previously found at MRG during 2018 (Barbaro et al., 2020) and with the summer mean concentration (30 ng m$^{-3}$) found at Sonnblick (Austria) and at Vallot (67-82 ng m$^{-3}$, French Alps) (Legrand et al., 2007). During winter, the reduced concentrations of total C2–C5 acids was also found by Legrand et al. (2007) in six sites along a west-east transect of 4000 km across Europe, considering rural and mountain sites. They found higher concentrations in winter when the surrounding local and regional sources were dominant while the summer concentrations came from larger-scale emissions.

The smaller organic acids are the most abundant: C2-oxalic acid shows a mean percentage of 36% all year-round, followed by C3-malonic (15%), C2-glycolic (14%), C2-formic (10%) and hC4-malic (7%) acids. The other carboxylic acids were found with percentage abundances <5%.

Carboxylic acids can be emitted by primary sources, such as biomass burning (Falkovich et al., 2005), fossil fuel combustion (Wang et al., 2006), vehicular exhaust (Kawamura and Kaplan, 1987) and marine input (Rinaldi et al., 2011), but these species are also produced by oxidation reactions in the atmosphere (Kawamura et al., 1996). Instead, pinonic and pinic acids are secondary products of α-pinene, the most important monoterpene released by biogenic sources. The quantification of the contribution of its sources is still poorly investigated. Kawamura et al. (2016) reviewed the main





sources and transformations while Feltracco et al. (2021) proposed a source apportionment for these species in Arctic
aerosol collected close to the sea.

Some diagnostic ratios of dicarboxylic acids can be used to define the photochemical aging of air masses. For example,
the C3-malonic acid to C4-succinic acid ratio is commonly used because C4-succinic acid can be degraded to C3-malonic
acid by decarboxylation reactions activated by OH radicals (Fu et al., 2013). At MRG, we found the C3-to-C4 ratio ranged
between 1 and 52, thus excluding primary emissions as the main sources. Typical values observed for vehicular emissions
ranged between 0.6 and 2.9 (Kawamura and Ikushima, 1993), while values obtained during an intense biomass burning
period were always below of 1 (Kundu et al., 2010). These data at MRG are consistent with previous values found during
the 2018 spring-summer campaign, where we hypnotized a photochemical production of these species at this mountain
site (Barbaro et al., 2020).

Another diagnostic ratio is the cis-C4-maleic acid to trans-C4-fumaric acid ratio because ambient photochemical
processes can affect their conversion. In our data, cis-C4-maleic acid concentrations are always higher than those of trans-
C4-fumaric acid, with a mean ratio of 3 (range 0.2-12), suggesting low photo-isomerization from cis-C4-maleic acid to
trans-C4-fumaric acid, and then low aerosol aging (Kawamura and Sakaguchi, 1999).

Oxidative processes are also involved in the formation of pinonic and pinic acid. These species showed higher total
concentration during spring ($6\pm3$ ng m$^{-3}$) and summer ($8\pm2$ ng m$^{-3}$ for 2021 and $7\pm2$ ng m$^{-3}$ for 2022) with respect to fall
($5\pm4$ ng m$^{-3}$) and winter ($4\pm2$ ng m$^{-3}$). These species reflect the vicinity of conifers as sources (Haque et al., 2016) and are
linked to the photo-oxidation processes in the atmosphere during the more active during summer period (Librando and
Tringali, 2005), as is also found for the other carboxylic acids.

### 3.2.3 Biomass burning tracers: anhydrosugars and phenolic compounds.

Biomass burning is an important primary source of many organic compounds and soot particulate matter. Several emission
sources can produce a biomass burning signal: wildfires, use of specific plant species as fuels, fossil fuel utilization, or
anthropogenic urban and industrial emissions from food preparation, and domestic heating (Simoneit, 2002). Each tree
species is constituted of cellulose, lignin and fillers. Cellulose is a long-chain, linear polymer of glucose and its burning
specifically produces levoglucosan and its isomers (mannosan and galactosan). Hemicelluloses are a mixture of
polysaccharides derived mainly from glucose, mannose, galactose, xylose, and arabinose, but the sugar composition
varies widely among different tree species. Finally, lignin is a biopolymer with vanillyl, syringyl and p-coumaryl moieties.
The study of the composition of phenolic compounds derived from lignin can be used to differentiate between the different
types of vegetation burned (Kuo et al., 2011).

Levoglucosan is one of the key tracers when investigating biomass burning contributions to aerosol (Simoneit, 1999). At
MRG, its concentration ranged between 0.1 and 48 ng m$^{-3}$ with an annual mean value of 2 ng m$^{-3}$. As shown in the Fig.
5, the annual trends of all the biomass burning tracers are deeply impacted by the Saharan Dust Event, recorded in March.
At MRG, excluding the samples affected by Saharan Dust, the mean values are $1.2\pm0.4$ ng m-3 and $1.3 \pm 0.9$ ng m$^{-3}$ for
summer 2021 and 2022, respectively, $1.2\pm0.9$ ng m$^{-3}$ for fall 2021, $1.6\pm0.7$ ng m$^{-3}$ for winter and $0.9\pm0.7$ ng m$^{-3}$ for spring
2022. Values below 10 ng m$^{-3}$ are consistent with high-level mountain sites such as Sonnblick (3105 m a.s.l., Austria),
with an annual mean concentration of 7.8 ng m$^{-3}$ (Puxbaum et al., 2007) or Alpe San Colombano (2250 m a.s.l., Italy)
where the values ranged between 5 and 10 ng m$^{-3}$ (Perrone et al., 2012). Considering the ratio between winter and summer
concentrations of levoglucosan, Puxbaum et al. (2007) found a relationship between this ratio and different sites, with
values of around 40 for continental rural sites and 3 for a maritime background site. At MRG, the winter-to-summer ratio
of 1.3 is aligned with elevated sites because low ratios can be explained by valley breeze effects, although it can be





assumed that there is considerable consumption of biomass fuels in the mountains. During winter, the mixing height in

alpine valleys is low and therefore the atmosphere at mountain peaks is effectively decoupled from the lower sites. In summer, elevated sites receive air masses from lower levels and the elevational gradient becomes less steep.

Levoglucosan and its isomer are usually major water soluble organic components in atmospheric aerosol impacted by wood smoke, but smoke aerosol can also be strongly enriched (by factors of 2–5) in the monosaccharides: glucose, arabinose, galactose and mannose (Medeiros et al., 2006). In particular, glucose is present at higher levels because it is

present in vascular plants, explaining the enrichment of those sugars in smoke-impacted aerosol (Cowie and Hedges, 1984). The samples collected at MRG in fall and winter are clearly impacted by biomass burning as shown in the panel related to the glucose-to-levoglucosan ratio in the Fig. 5. In fact, Medeiros et al. (Medeiros et al., 2006) reported a value of about 4.5 of glucose/levoglucosan ratio for smoke-free samples, and considerably lower values of 0.9 for smoke samples. On the contrary, during spring and summer the ratio went beyond the smoke-free threshold, indicating a natural

input as most probable source. It is plausible that this is also due to domestic heating with wood in the mountain areas.

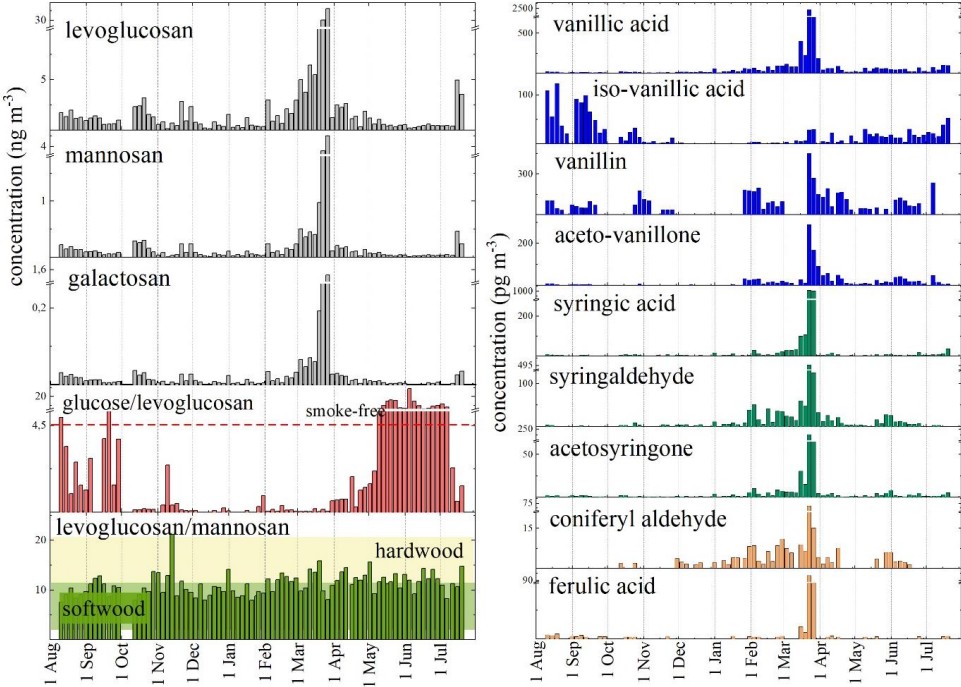

**Figure 5. Seasonal trend of concentrations of levoglucosan and its isomers and phenolic compounds. Two Discriminator ratios are reported in two panels: glucose-to-levoglucosan with the threshold of smoke free samples (Medeiros et al., 2006) and levoglucosan-to-mannosan with the range of different biomass fuels (Fabbri et al., 2009).**


The diagnostic ratio between levoglucosan and mannosan can also help to define the type of vegetation burned. Fabbri et al. (2009) proposed this discrimination ratio after analyzing different type of plants and their levoglucosan-to-mannosan ratios. Ratios of 13.8-22 and 0.6-13.8 are indicative of hardwood and softwood combustion, respectively, while values > 24 indicate grass fires. Figure 5 shows that at MRG softwood is the main combustion fuel. This is also confirmed by the

main presence of vanillyl species such as vanillic acid, vanillin, iso-vanilllic acid, aceto-vanillone, typical compounds present in softwood (Fig. 5). An increasing concentration of all types of phenolic compounds is found in the samples



affected by Saharan Dust, suggesting a mix of burnt vegetation. The increase in all biomass burning tracers in the Saharan Dust particles is due to atmospheric long range transport (Fig. S2) of air masses over Spain and France, where domestic heating with wood (and coal) in small private stoves can produce this type of signal. The only phenolic compound without

a peak related to Saharan Dust is iso-vanillic acid and its trend is similar to those of the organic acids (Fig. 5). This suggests a photochemical source for this compound, that is not correlated with biomass burning sources.

### 3.2.4 Biogenic aerosol: saccharides, alcohol -sugars and free amino acids

Primary biological aerosol particles (PBAPs) can be defined as solid airborne particles that are directly emitted by the biosphere into the atmosphere (Després et al., 2012). PBAPs are likely to be a major source of proteinaceous materials in

the atmosphere (Matos et al., 2016). They can be emitted as biological aerosols such as virus, algae, fungi, bacteria, spores and pollen, fragments of plants and insects, they also can be associated with anthropogenic sources such as industry, agricultural practices, and wastewater treatment plants (Lazaridis, 2008).

Free amino acids and saccharides have been traditionally used for the identification of particles of biological origin (Bauer et al., 2008; Matos et al., 2016; Ruiz-Jimenez et al., 2021). Seventeen free L- and D-amino acids (L-Ala, D-Ala, L-Arg,

L-Asn, L-Asp, D-Asp, Gly, L-Glu, L-Leu, L-Hys, L-Hyp, L-Phe, D-Phe, L-Pro, L-Thr, L-Tyr, L-Val) were found in the MRG samples (Fig. 6). The total mean concentrations were $4\pm2$ and $5\pm1$ ng m$^{-3}$ in summer 2021 and 2022 respectively, $0.9\pm0.7$ ng m$^{-3}$ in fall, $0.6\pm0.8$ ng m$^{-3}$ in winter, and $3\pm1$ ng m$^{-3}$ in spring. As also shown in Fig. 6, are the higher concentrations of L and D- free amino acids that were found in summer, while only L- free amino acids were abundant in the spring, D-amino acids increased in concentration in the late spring. The concentration values are consistent with

those found in PM$_{10}$ in the Arctic (Feltracco et al., 2021a; Scalabrin et al., 2012) and in the Antarctic (Barbaro et al., 2015a; Feltracco et al., 2023), but lower than typical values found in rural (Mace, 2003) or urban sites (Di Filippo et al., 2014; Zangrando et al., 2016).





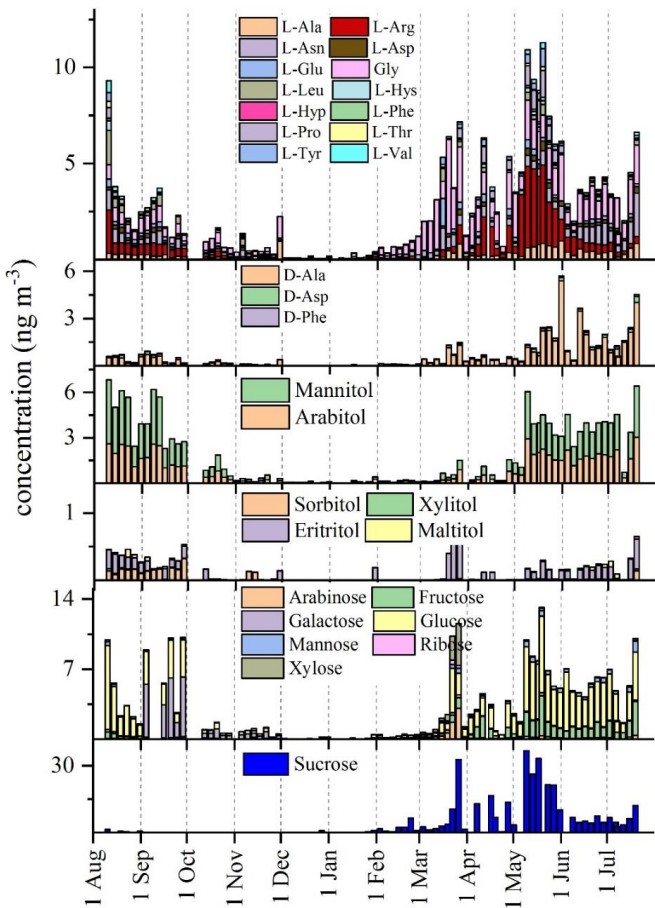

**Figure 6. Seasonal variations of L- and D- free amino acids and sugars in the aerosol collected at MRG from August 2021 and the end of July 2022.**

Seven monosaccharides (arabinose, fructose, galactose, glucose, mannose, ribose and xylose), six alcohol-sugars (arabitol, mannitol, ribitol, iso-erythriol, sorbitol, and xylitol) and one disaccharide, sucrose, were found in the MRG aerosol from August 2021 to the end of July 2022. The highest concentrations were determined in spring and summer (Fig. 6), with the total sugars mean concentrations of $9\pm4$ ng m$^{-3}$ and $14\pm8$ ng m$^{-3}$ for summers 2021 and 2022 respectively, $17\pm14$ ng m$^{-3}$ for spring, while concentration fell to $2\pm3$ ng m$^{-3}$ and $1\pm2$ ng m$^{-3}$ in fall and winter respectively. Mannitol and arabitol are shown in another panel of Fig. 6 because these species are specific tracers of fungal spores (Bauer et al., 2008) and they seems to have another trend with significant concentrations from May to October. Glucose and other monosaccharides seem to be weakly affected by the Saharan Dust while other biogenic species (free amino acids and sugars, Fig. 6) are not affected by the Saharan Dust Event at all, suggesting that long-range transport should be a negligible source compared to local emissions in the mountain environment.



### 3.2.5 Elemental composition and possible tracer of aerosol sources

The use of trace element enrichment factors (EFs) is useful to highlight the contribution of non-natural or anthropogenic sources on elemental concentrations. An EF is calculated as follows:

$$EF_i = (i/j)_{atmosphere} / (i/j)_{upper\ crust}$$

where $EF_i$ is the enrichment factor of element i, j is a reference element of crustal origin, $(i/j)_{atmosphere}$ is the ratio of element i to element j in the atmosphere and $(i/j)_{upper\ crust}$ is the ratio of element i to element j in the upper crust (Wedepohl, 1995). We used Al as the reference element in our EF calculations following the suggestion of Gao et al. (1992). A crustal derived element (geogenic) shows an EF near 1, while significantly higher EF values indicate a non-

geogenic (not crustal-derived) element: EF between 10 and 100 indicates a moderate enrichment while an EF above 100 indicates a prevalent not-crustal origin.

Similarly, we can calculate the marine enrichment factors (MEFs) relative to sea water concentrations (abundance in sea water from Nozaki, 2010) to highlight a possible marine source. MEFs were calculated as:

$$MEF_i = (i/j)_{atmosphere} / (i/j)_{sea\ water}$$

ssCa (sea salt derived Ca) was used as reference element (j) of marine origin. A MEF <1 indicates a marine derived element, while higher values indicate a moderate enrichment (10-100) and high values (>100) relative to sea water, exclude the prevalence of marine sources.

On the basis of the EF values (Fig. S6a), we can hypothesize that the prevalent source for K, Na, Cs, Sr, Ti, Ca, V, Ba, Li, Mn, Rb, Co, Tl, Fe, Mg and LREE is geogenic with some exceptions during fall and winter seasons when a moderate

enrichment of these elements is evident. MREE, HREE, and U show a high variability between geogenic and non-geogenic sources especially from fall 2021 to spring 2022. Cr, Cu, Pb, Zn, Ni, Sb and Cd show a prevalently non geogenic source during all the seasons, but not exclusively, especially during the two summer periods and the winter one. Ag and Mo show always a clear non geogenic source. The MEF values (Fig. S6b) allow us to see that Na, K, Sr, Mg, Ca, are partially related to a marine source. All other elements show a value higher than 1 indicating a prevalently non- marine

source.

To better discriminate between the possible sources, we evaluated various elements by considering their ability to characterize certain sources. The principal sources of MRG aerosol are a local crustal contribution, long range transport of Saharan dust, sea-salt spray from the Mediterranean basin, biomass combustion, either from fires or from domestic heating, and to a lesser extent, traffic, refinery emissions, and oil combustion processes.

To do a first survey of sources other than crustal ones that could characterize the MRG samples, we plotted a ternary diagram of La-Ce-Sm (Fig. 7), three rare earth elements that show a clear geogenic origin based on their EF values. Using this method, all seasons show different sources other than the earth's crust, although it is possible to recognize a prevalently crustal origin in the fall and winter samples, even if not exclusively so. In the spring and summer samples sources are prevalently different to the crust.





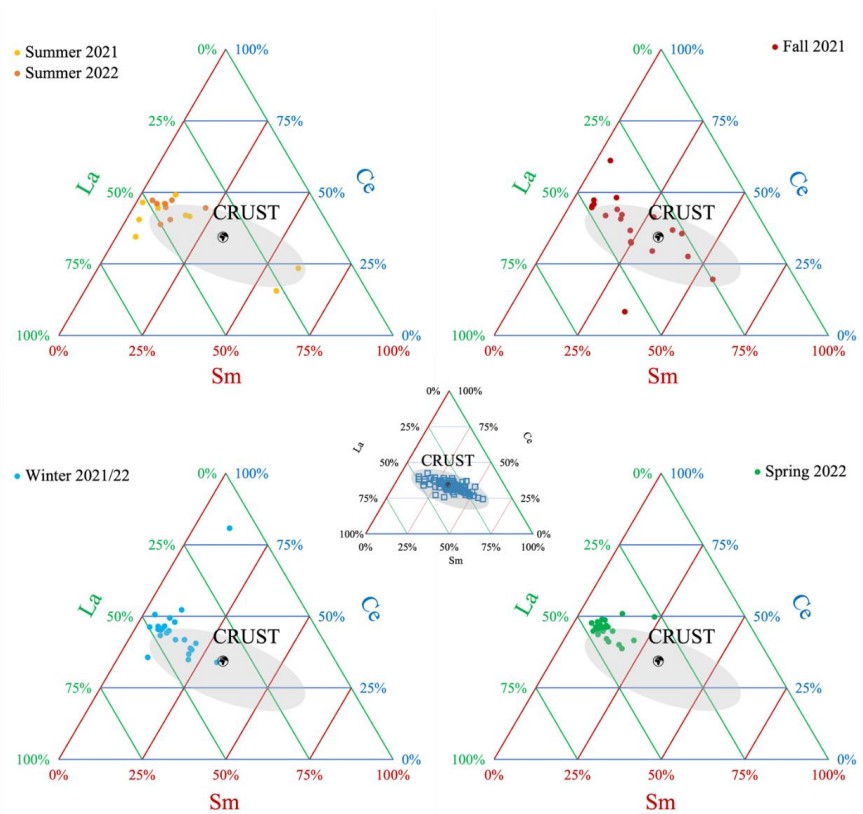


**Figure 7. Ternary diagrams La-Ce-Sm. La and Sm were multiplied by 2 and 10 respectively to have the upper continental crust composition at the center of the graph. The grey ellipse represents the crustal composition of the area around MRG station (data from Lustrino et al. 2019 and references therein).**

Due to the high elevation of MRG station, we firstly considered possible origins from natural sources after both local and long-range transport. EF factors (Fig. S6) indicate some elements as having prevalent, but not exclusively, geogenic origins. In particular, we have focused our attention on the Mg distribution, whose source could either be the crust and/or sea-salt spray. The comparison between the water-soluble fraction and the total content of Mg reveals the two possible origins of this element. In the Fig. 8a we compare the distribution of water-soluble Mg (Mg2+, light green line) and total

Mg (Mg, green line) normalised to the mass of $PM_{10}$ collected. Mg of a geogenic (crustal) origin is expected to be mostly non-water soluble while Mg from sea-salt spray is mainly related to the water-soluble fraction. Due to the different concentrations, the peaks of Mg2+ in Fig. 8a are not very evident, the same data are reported in Fig. 8b and c where the scale differences make the variances in the signal more obvious.

In Fig. 8a and 8b various Mg peaks of both the soluble and total fractions, can be recognized. During fall 2021 some

peaks of Mg2+ and Mg are evident, while in late winter a peak of total Mg is visible in correspondence with the Saharan Dust event, but the corresponding Mg2+ value is very low. Considering that Mg is one of the major components of sea-salt spray and is also present at very high levels in Saharan dust we can use the Mg fractions as specific tracers, in Mediterranean region, of these two different sources.





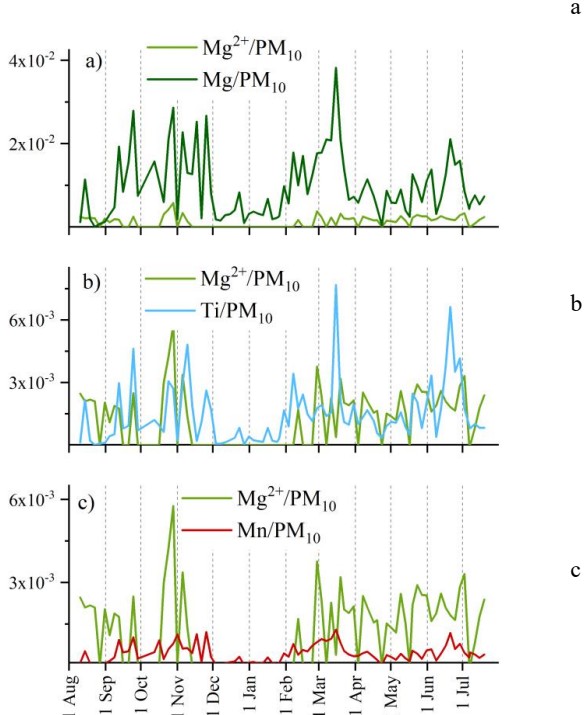

**Figure 8. Comparison between water soluble fraction of Mg and total fraction of Mg (a), total Ti (b), total Mn (c). Mg, Ti and Mn are normalized using PM$_{10}$ mass concentration.**

Mg from sea salt origin, has in fact, a high solubility while Mg from Saharan dusts, originating from clay minerals, has a low solubility (Perrino et al., 2008). Based on this, we can attribute the Fall 2021 peaks to a marine origin while the late-winter peak seems to be related to a Saharan event, also considering that during Saharan events there is a high dust loading (Fig. 1). In Fig. 8 b and c the distribution of total Ti and total Mn, respectively, are compared with the water-soluble fraction of Mg to try and determine the possible source of the late-winter PM$_{10}$ peak. Ti and Mn are recognized as indicators of Saharan dust (Perrino et al., 2008); the observed distributions in Fig. 8b and c confirm the evidence of the Saharan input, already highlighted by the Mg distribution.

In Fig. 9, Ti, Mn and Mo, a third element related to the North African area (Wong et al., 2020a), are plotted highlighting samples that could be completely or, at least, partially related to a North African source. Only one sample can be recognized as fully related to a Saharan event, the spring one (blue dot in Fig. 9). Also, two other spring samples (green dots) could also be partially related to possible minor events characterized by a North African component or by a component coming from the route followed by Saharan dusts passing through Spain and France before reaching the Italian Alps.



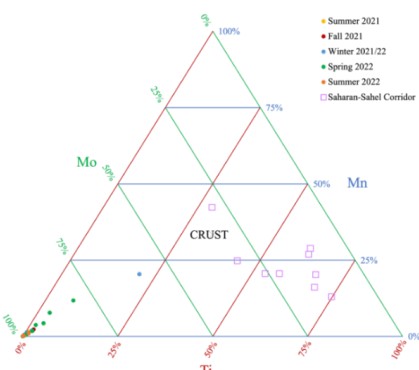

**Figure 9. Ternary diagrams: Mo-Mn-Ti. Ti and Mn were multiplied by 0.00045 and 0.00265 respectively to have the upper continental crust composition at the center of the graph. Data of Saharan-Sahel Corridors from Moreno et al (2006).**

Other sources to be considered when characterizing aerosol samples are for example, biomass burning, both from fire and domestic uses. To highlight the possible contribution of this kind of source to the MRG aerosol composition, we plotted three rare earth elements Eu-Ho-Yb in a ternary diagram (Fig. 10). These elements give information on fire events, and can distinguish between different biomass ashes: in particular, it is possible to discriminate hardwood and softwood fires from grass and shrub fires (Dukes et al., 2018). In Fig. 10 we show the sample distributions; in comparison with literature

data for tree and biomass ashes as reported in (Flood, 2019; Perämäki et al., 2019; Vassilev and Vassileva, 2020).
The fall samples, red dots, and some spring samples, green dots, are in the tree ash region, indicating hardwood and softwood burning as a possible source for PM, probably coming from domestic heating with wood (and coal), that it is known to producing this type of signal in aerosol. Some samples are in the shrub area and in the region between bare ground and shrub (Dukes et al., 2018). In the spring and summer of 2022 there were high temperatures, severe droughts,

and many fires, especially in Spain. So, the samples in the area between bare ground and shrubs, in Fig. 10, may be linked to these wildfire events. In particular, the spring samples, the green dots, in this region are the same samples that were already highlighted in Fig. 10 and hypothesized as related to minor Saharan events or to the Spain-France region. On this basis of these results, the samples can be linked to fire events instead of Saharan events.

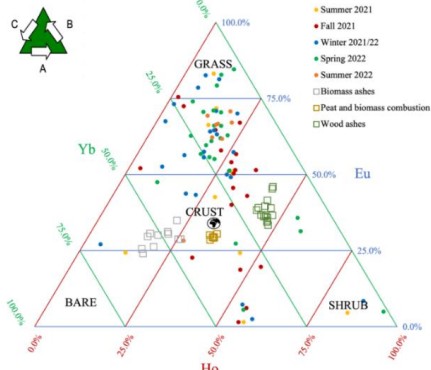

**Figure 10. Ternary diagram of Eu-Ho-Yb. Eu and Yb were multiplied by 0.64 and 0.41 respectively to have the upper continental crust composition at the centre of the graph.**



We plotted the MRG aerosol samples in a ternary diagram based on V, La and Ce with the aim of recognizing aerosol from three possible anthropogenic sources related to refinery emissions, oil combustion processes and vehicular traffic emissions (Fig. 11).

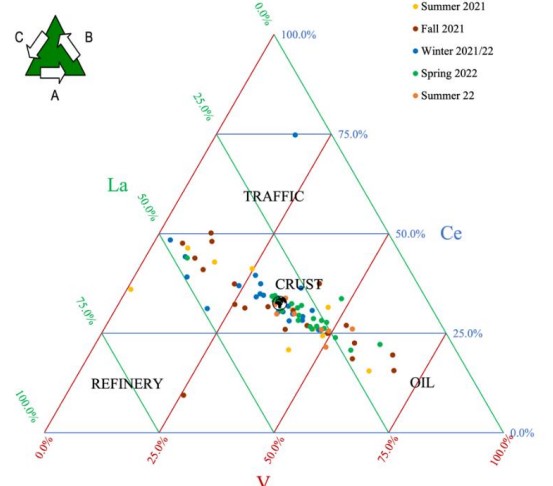

**Figure 11. V-La-Ce ternary graph. La and Ce were multiplied by 3.1 and 1.54 respectively to have the upper continental crust composition at the center of the graph. Indication of oil, traffic, and refinery corners from (Moreno et al., 2008).**

Samples are mainly distributed in the central part of the diagram, around the crustal composition, indicating a prevalently crustal source for V. It is also possible to see, however, a group of samples shifting toward higher values of Ce an La and another group toward higher values of V. These two different groups, seem to indicate possibly different anthropogenic sources: namely traffic/refinery and oil combustion processes respectively.

### 3.3 Source apportionment

A source apportionment approach using Positive Matrix Factorization yielded a reasonable resolution employing four factors. To ascertain the most suitable number of factors possessing meaningful physical implications, an evaluation was conducted using parameters IM (representing the maximum individual column mean) and IS (indicating the maximum individual column standard deviation), both derived from the scaled residual matrix. This analysis was supplemented by considering Q-values (a parameter assessing the goodness of fit) (Viana et al., 2008). The analysis of the scaled residual is symmetrically distributed for almost all variables, meaning that the model is able to fit quite well each chemical species. Considering the plotting of modelled data obtained by PMF and the observed data, $R^2$ values often above 0.9 was obtained for the considered species (Table S1). Some exceptions are recognized: for example, levoglucosan, the key tracer of biomass burning, is well reconstructed during the entire sampling period but its high concentration during the intense Saharan Dust event of March is not recognized, probably because this concentration is not correlated with the Saharan source but to transport over contaminated areas (Spain and France). In general, the model is able to reasonably reconstruct the observed concentrations with a slope of 0.977 and an $R^2$ of 0.97 considering the entire dataset, but the concentrations related to the intense Saharan Dust event of March forced the linearity. Considering the dataset without this point, the linearity is preserved with a slope of 0.897 and an $R^2$ of 0.84.



In Fig. 12, the profiles of each factor are reported in terms of absolute and relative concentrations, with the relative
contribution of the factors. The error bars represent the standard deviations of the Bootstrap runs. The Fig. 13 reports the
contribution in term of concentrations of each factor considering the entire sampling period. Moreover, to define the
source of these contributions, a comparison with air mass back trajectories is also shown in terms of percentage calculated
by cluster means obtained for each sample (Fig. S7).

The first factor describes 41% of the PM$_{10}$ and it is mainly characterized by crustal elements, such as Mg and Ca but also
some trace elements and rare earth elements (Fig. 12). This attribution is also confirmed by the presence of air masses
coming from North Africa, also defined using back-trajectories. Although the most intense Saharan Dust event is
recognized in the sample from 15-19 March, zooming into the factor contributions, by putting a break in the y-axes,
reveals the other Saharan Dust events that were detectable at Col Margherita observatory.

The second contributed accounted for only 11% of the total PM$_{10}$ concentration and it mainly loads with Mo and U,
although there are also some rare earth elements (Ho and Yb), typically associated with atmospheric long-range transport,
having a percentage above 25% of the total PM$_{10}$ concentration (Fig. 12). Considering the back-trajectories (Fig. 13 and
S8), the annual trend of contributions of this factor seems to be related to the intrusion of air masses coming from the
Atlantic Ocean and the Atlantic Ocean across to North America. Coupling this evidence with our knowledge about Mo
and U possible origins (Aciego et al., 2015; Flett et al., 2021; Masson et al., 2015; Qi et al., 2016; Wong et al., 2021,
2020b) and with the EF's indications, we can hypothesize a long-range transport from polluted areas with a further
enrichment when oceanic air masses pass over the industrialized areas of central-northern Europe.

The third factor (Fig. 12), accounting for 29% of the PM$_{10}$ concentration, was identified as biogenic sources because it is
characterized by the major ions of saline components and water-soluble organic compounds, such as carboxylic acids
(CA), L- and D- free amino acids (L-FAA and D-FAA), mannitol, key tracer of fungal spores, glucose, and sugars with
a typical biogenic input. The seasonal trend of this factor (Fig. 13) showed higher concentrations during the warm season,
linked to an increase in biogenic activities, but also to seasonal effects due to 1) valley breeze and 2) planetary boundary
layer (PBL) changes. Previous studies at MRG already demonstrated the increase in PM$_{10}$ and mercury concentrations
during summer (Feltracco et al., 2022; Vardè et al., 2022).

Finally, the fourth factor, which describes 19% of PM$_{10}$, is mainly defined by levoglucosan and phenolic compounds,
suggesting biomass burning sources, and some heavy metals such as Pb, V and Ag, adding an anthropogenic source (Fig.
12). This factor showed the highest concentrations between February and April when the air masses mainly come from
Europe without any marine input. The correspondence of contributions of factor 4 with air masses clearly suggests the
impact of urban areas on air quality.





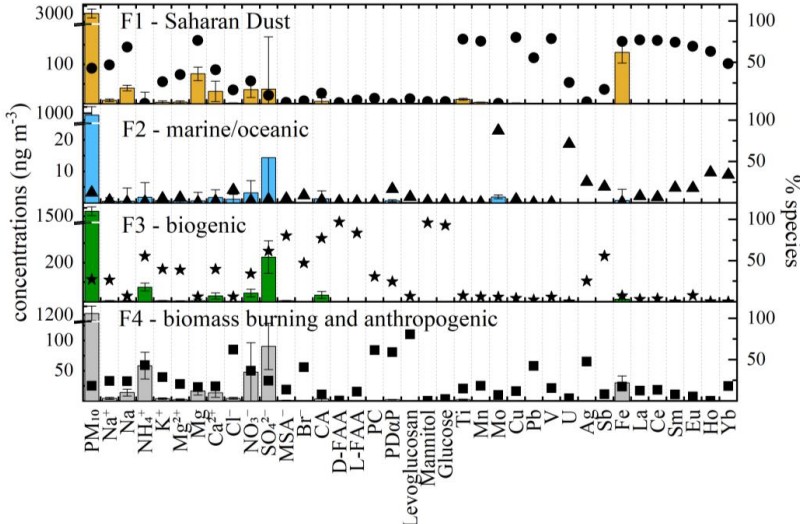

Figure 12. Source profiles obtained with the PMF. The bars identify the species in terms of concentrations that mainly characterize each factor profile. Error bars were obtained with bootstrap method. The points describe the species in terms of relative concentrations. (PC are phenolic compounds, CA are carboxylic acids, FAA are free amino acids, PDαP are photodegradation products of α-pinene).

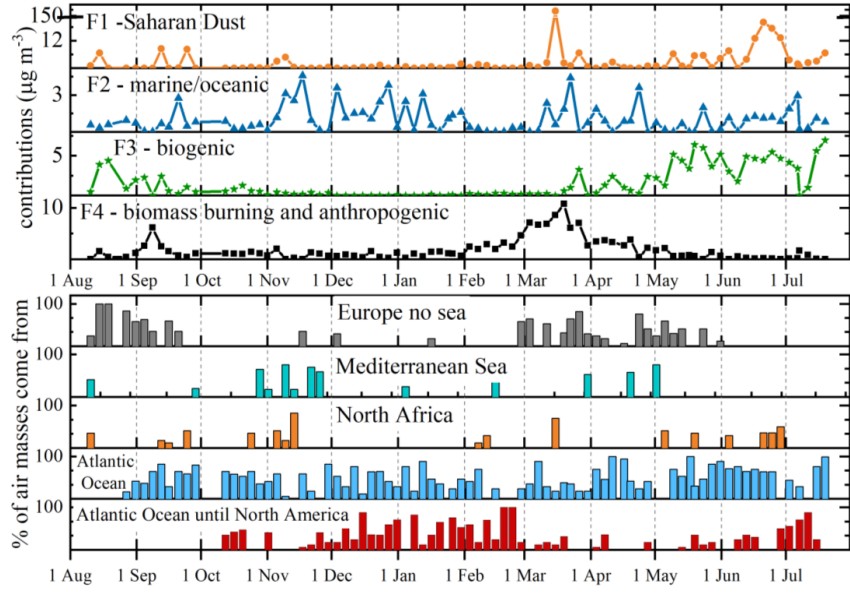

Figure 13. Factor contributions during the sampling period is reported in the Fig. above, compared with the % of air masses obtained by each back-trajectory reported in Fig. S8.



## 4. Conclusions

The chemical composition of atmospheric aerosol from an entire year (August 2021 - July 2022) was carried out at the Col Margherita Atmospheric Observatory, high-altitude background site in the Eastern Italian Alps. For the first time more than one hundred chemical markers from a whole years' worth of samples were determined, including major ions, organic acids, sugars, free amino acids, trace elements and rare earth elements.

The main aims of this study were to define i) the main sources of aerosol arriving at this remote site, ii) background values
for the chemical markers when describing the aerosol inputs to this remote area. To the best of our knowledge, information on high mountain aerosol is limited to major ions, some trace elements, or some organic compounds.

For the first time, intrusions of air masses impacted by Saharan Dust events are demonstrated and chemically characterized. Each class of analytes are discussed, considering these chemical species as specific markers of sources or processes. Sulfate and nitrate are the most abundant species for the entire sampling period, but the crustal elements (Ca,
Mg and Fe) are important component of alpine aerosol. Using specific diagnostic ratios, the anthropogenic inputs of sulphate and nitrate were demonstrated. Photochemical processes are also recognized using carboxylic acids and photo-oxidation products of α-pinene, which showed high concentrations when irradiation was higher.

Levoglucosan and phenolic compounds have allowed us to define the different types of biomass burning that occurred, distinguishing the contribution come from domestic heating in European cities and that coming from air masses
transported though the Saharan Dust events. The biogenic input was characterized by using some water-soluble organic compounds, such as free amino acids and sugars. For example, the presence of specific tracers of fungal spores such as mannitol and arabitol are mainly recognized during spring and summer, as clearly expected.

Enrichment factors (EFs) for trace elements have proven invaluable in emphasizing the impact of sources beyond natural origins on elemental concentrations.

The dominant origin for elements like K, Na, Cs, Sr, Ti, Ca, V, Ba, Li, Mn, Rb, Co, Tl, Fe, Mg, and LREE is geogenic, except during fall and winter seasons when a moderate increase in these elements is observable. On the other hand, rare earth elements and U exhibit significant fluctuations between geogenic and non-geogenic sources, particularly from fall 2021 to spring 2022. Elements such as Cr, Cu, Pb, Zn, Ni, Sb, and Cd primarily stem from non-geogenic sources across all seasons, although with occasional deviations, notably during both summer periods and winter phases. Ag and Mo
consistently indicate a clear non-geogenic origin.

To sum up, a source apportionment using Positive Matrix Factor was performed, obtaining four factors: 1) Saharan Dust Events; 2) long range marine/anthropogenic input; 3) biogenic sources; 4) biomass burning and anthropogenic emissions. We can conclude that, despite the remoteness of the Col Margherita site, anthropogenic pollution, both from regional and long-range transport, impacts this area.

**Competing interest**

The contact author has declared that none of the authors has any competing interests.



**Acknowledgments**

This study was carried out within the "Interconnected Nord-Est Innovation Ecosystem (iNEST)" project and received funding from the European Union Next-GenerationEU - National Recovery and Resilience Plan (NRRP) – MISSION 4 COMPONENT 2, INVESTIMENT N. ECS00000043 – CUP N. H43C22000540006. This manuscript reflects only the authors' views and opinions, neither the European Union nor the European Commission can be considered responsible for them.

This work has benefited from the infrastructural support of the Centre for Trace Analysis (CeTrA) of Ca' Foscari University through the project IR0000032 – ITINERIS, Italian Integrated Environmental Research Infrastructures System, funded by EU - Next Generation EU, PNRR- Mission 4 "Education and Research" - Component 2: "From research to business" - Investment 3.1: "Fund for the realisation of an integrated system of research and innovation infrastructures".

The results are produced also in the framework "DECIPHER - Disentangling mechanisms controlling atmospheric transport and mixing processes over mountain areas at different space and timescales (2022NEWP4J) funded by PRIN MUR.

The authors thank Elga Lab water, High Wycombe UK for supplying the pure water systems used in this study. A special thanks to the personnel of the "Ski area San Pellegrino", in particular the director Renzo Minella, the head of the cable

car service and the technicians their invaluable help and cooperation during the sampling activities at the Observatory.

**Authors statement**

Elena Barbaro: Conceptualization, Investigation, Formal analysis, Data curation, Writing - original draft, Writing - review & editing. Matteo Feltracco: Conceptualization, Investigation, Formal analysis, Data curation, Writing - original draft

Fabrizio De Blasi: Investigation, Data curation, Writing - original draft. Clara Turetta: Investigation, Formal analysis, Data curation, Writing - original draft, Writing - review & editing. Marta Radaelli: Formal analysis, Writing - review & editing. Warren Cairns: Investigation, Writing - review & editing. Giulio Cozzi: Investigation, Writing - review & editing. Giovanna Mazzi: Investigation, Writing - review & editing. Marco Casula: Formal analysis, Writing - review & editing. Jacopo Gabrieli: Investigation, Supervision, Writing - review & editing. Carlo Barbante: Investigation, Supervision,

Writing - review & editing. Andrea Gambaro: Investigation, Supervision, Writing - review & editing.



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
