# Peer review of "Chemical characterization of atmospheric aerosols at a highaltitude mountain site: a study of source apportionment"

_EGUsphere, 2023_

## Referee Comment (RC2)

This study presents a thorough discussion on the chemical composition of $PM_{10}$ at Col Margherita. The datasets and analysis are robust and provide valuable insights for high-altitude background sites. The source apportionment results are interesting, as mass trajectories from different regions have distinct chemical characteristics. The manuscript can be further improved in the following aspects:

1. The abstract only included the meaning of the study and the method. The main conclusions were not mentioned.
2. It seems only ~30 compounds were used in the PMF analysis, despite that ~100 is measured. What's the reason? What would happen like if all the measured species are included?
3. Figure 13, the PMF results only show contributions from primary sources and are mostly transported from other regions, are there secondary sources or background aerosols?
4. Section 3.2 provides too many details and some may not be relevant to the main conclusions of the manuscript, these parts can go to the supplementary information. For instance, Figure 7-11 provides too many information on the trace elements and it distract the readers a lot.
5. Line 468, the traffic/refinery and oil combustion process are not recognized in PMF. Please discuss the reasons.

Other minor comments:
1. Line 57-62 was duplicated with Line 165-173.
2. Line 94: what instrument is used to analyze the organic compounds?
3. The definition of periods are obscure, please define in the method which month are regarded as "spring", "summer", "autumn", "winter", "the seasons without snow cover", "the snow season", "the cold period".
4. Figure 2, what is "CA", why it increases in Spring and Summer?
5. Line 161-162 please give a brief preview here of what might be missing in December and January, and mention the exact section that discuss this in detail. It's hard for the readers to search for the clues in the long discussion part.
6. Line 173, please give a brief ratio of the concentration measured by the two methods.
7. Line 266-268 The logic here is hard to understand. Authors say that the two acids are from α-pinene, then they say its poorly understood? And the following examples in Line 268-230 should provide specific conclusions of the two studies.

---

## Author Response (AR1)

**Anonymous Referee #1,**

The paper deals with analysis of one-year long dataset of very comprehensive chemical analysis results of PM10 samples at high altitude mountain sampling site at the Col Margherita Atmospheric Observatory. The authors obtained a unique dataset of 100 inorganic and organic chemical species based on analysis of 87 four-day samples using low volume samplers and a flow rate 38.3 l/min. Elemental analysis was provided using ICP-SFMS, water soluble compounds were analysed for inorganic ions, organic acids, monosaccharides, sugar alcohols, anhydrosugars, sucrose, amino acids, phenolic compounds and photooxidation products of α-pinen. Finally, the data were analysed using different statistical methods including PMF and air mass trajectory analysis. However, there are major issues that prevent to publish the manuscript in current state.

*A: Thanks to Anonymous Referee #1 for the suggestion in his/her revision. We have modified the manuscript using the suggestions to improve its quality.*

Major issues

Although, the reviewer cannot fully accept using balances with 0.1 mg resolution for gravimetry of samples taken using low volume sampler in such pristine environment despite four day samples, this is not a major objection.

*A: We totally agree with reviewer, because it is great mistake. We check the balance used and also the model was wrong. All data was collected with a readability [d] of 0,00001 g. Our gravimetric analysis was performed using a Sartorius CP225D balance (precision ± 0.01 mg), placed inside a clean room class 1000, at Ca' Foscari University, Venice, as also described by Gregoris et al. (2021; https://doi.org/10.1016/j.apr.2020.11.007). Thanks for the comment.*

More important is that except for elemental analysis no other method used is even named and at least some parameters of analysis should be mentioned not only references that are given here.

*A: As suggested by referee, we added the description of each method using to produce the dataset: "Briefly, an ion chromatograph (IC, Thermo Scientific DionexTM ICS-5000, Waltham, MA, USA) coupled with a single quadrupole mass spectrometer (MSQ PlusTM, Thermo Scientific, Bremen, Germany) was used to analyse anionic compounds: Cl-, NO3- , SO42- , Br-, methanesulphonate and carboxylic acids. The separation was carried out using an anion exchange column Dionex Ion AS19 2 x 250 mm equipped with Dionex Ion Pac AG19 guard column (2 x 50mm) using sodium hydroxide (NaOH), produced by an eluent generator, as mobile phase and with with a flow of 0.25 mL min-1. Cationic species (Na+, NH4+, Mg2+, Ca2+, K+) were determined using a capillary IC (Thermo Scientific Dionex ICS-5000), equipped with a capillary cation exchange column (Dionex IonPac CS19-4 µm, 0.4 × 250 mm), a guard column (Dionex IonPac CG19-4 µm, 0.4× 50mm) connected with a conductibility detector (Barbaro et al., 2020).*

*The determination of sugars (Barbaro et al., 2015b) in aerosol samples involved the utilization of the same ion chromatograph with quadrupole mass spectrometer (ICS-5000 and MSQ Plus™, Thermo Scientific). The separation of six monosaccharides (arabinose, fructose, galactose, glucose, mannose, ribose, xylose), sucrose, and maltitol was achieved using a CarboPac PA10™ column (Thermo Scientific, 2 × 250 mm) equipped with a CarboPac PA10™ guard column (2 × 50 mm). Separation of seven alcohol-sugars (arabitol, erythritol, mannitol, ribitol, sorbitol, xylitol, galactitol) and anhydrosugars (levoglucosan, mannosan, and galactosan) was accomplished using a CarboPac MA1™ analytical column (Thermo Scientific, 2 × 250 mm) with an AminoTrap column (2 × 50 mm). In both methods, the injection volume was 50 µL, and the flow rate of NaOH was 0.25 mL min−1.*

*Compound detection was performed in the selected ion monitoring (SIM) acquisition mode in negative electrospray ionization (−ESI-MS).*

*The determination of free L- and D-amino acids was conducted using an Agilent 1100 Series HPLC System (Waldbronn, Germany) featuring a binary pump, vacuum degasser, and autosampler. This system was coupled with an API 4000 Triple Quadrupole Mass Spectrometer (Applied Biosystem/MSD SCIEX, Concord, Ontario, Canada) employing a TurboV electrospray source in positive mode with multiple reaction monitoring (MRM). The separation of free L- and D-amino acids utilized a chiral 2.1 × 250 mm CHIROBIOTIC TAG column (Advanced Separation Technologies Inc, USA) with ultrapure water containing 0.1% formic acid (eluent A) and methanol with 0.1% formic acid (eluent B) as mobile phases. The binary elution program at a flow rate of 0.15 mL min−1 and the injection volume was 100 μL (Barbaro et al., 2015a).*

*For the determination of phenolic compounds (vanillic (VA), isovanillic (IVA), homovanillic (HA), syringic (SyA), p-coumaric (PA), ferulic acids (FA), vanillin (VAN), syringaldehyde (SyAH), coniferyl aldehyde (CAH), acetosyringone (SyAC), acetovanillone (VAC)) and photo-oxidation products of α-pinene (cis-pinonic acid and pinic acid), the same HPLC-MS/MS system used for amino acids determination was employed. Both separations used a Zorbax Extend C18 (150 mm × 4.6 mm, 3.5 μm, Agilent) column with a 0.01% formic acid aqueous solution (eluent A) and a solution of methanol/acetonitrile 80/20 (eluent B) as mobile phases, and a flow rate of 0.5 mL min−1. The injection volume was 100 μL, and the ESI source operated in negative mode. The binary elution programs for phenolic compounds and pinonic and pinic acids were reported in Zangrando et al. (2013) and Feltracco et al. (2018)."*

Second, based on average seasonal composition of Fig 2 and Fig 3 water soluble anions (mainly sulphates and nitrates) form important mass of PM10 mass and based on ionic balance Fig 3 they are compensated mainly as usual by NH4+ but in similar way by Ca2+. It is clear that in such case NH4+/SO42- ratio cannot be used as a measure for nitrates formation, because large part of pH control is provided by Ca2+ and other cations, not only by NH4+. In this case, 96 hour sampling also prevent such conclusion as, due to long sampling, there is large probability that filter pass through high RH phase when part of aerosol become liquid and all redundant NH4+ will be removed from the solution into gas phase by Ca2+ and other cations especially when they are present as carbonates as the authors suppose. In the same time, free nitric acid can be captured on the filter if available in the air. Based on the above, the lines 230-245 should be removed.

*A: As suggested by Anonymous Referee #1, we removed the lines 230-245 and the figure 4.*

Third, source apportionment needs substantial improvement. The matrix itself is already well below recommended threshold ratio between number of lines to number of columns that is 3:1.

*A: Henry et al. (1984) in their "Reviews of receptor model fundamentals" reports this equation to calculate the number of cases (samples): N>30 + ((V+3)/2), where N is the number of cases (samples) and V is the number of variables (number of chemical species). The Positive Matrix Factorization (PMF) model is based on this formula when we consider the sufficient number of samples. In our specific cases we have 85 samples(N) which are major than (30+((37+3)/2))=50.*

This could be improved by removing several apparently quite correlated crustal related species. The method used for selection of number of factors is not referenced and it is not clear if any of the used chemical species was set "weak" or "bad".

*A: We agree with reviewer, and we improved the PMF section as follows: "All variables are considered "strong", except for photodegradation products of α-pinene which are considered "weak" because their signal-to-noise ratios were lower than two."*

Moreover, it is quite clear on the base of presented solution, that uncertainty matrix is not constructed properly, the whole solution is driven by elemental composition and major species among water soluble ions, amino acids, PDαP, PC or sugar-like compounds are modelled quite badly (Table S1). Most probably the errors of elemental composition are set much lower in comparison to rest of the data, harmonisation of uncertainties is needed.

*A: We calculated the uncertainty matrix as described in the guide: "The equation-based uncertainty file provides species-specific parameters that EPA PMF 5.0 uses to calculate uncertainties for each sample. [...] If the concentration is less than or equal to the MDL provided, the uncertainty (Unc) is calculated using a fixed fraction of the MDL (Equation 5-1; Polissar et al., 1998).*

$$Unc = (5/6) * MDL.$$

*If the concentration is greater than the MDL provided, the calculation is based on a user provided fraction of the concentration and MDL (Equation 5-2).*

$$Unc = \sqrt{(error\ fraction * conc)^2 + (0.5 * MDL)^2}$$

*We cannot modify the errors because these values are real, calculated by validation procedures while MDLs are calculated as 3 times the standard deviation of field blank.*

*However, we agree with reviewers that in the main manuscript was not properly reported how these values were calculated. So, we improved it as follows: "The uncertainties of each sample concentration were defined using the relative standard deviations for each variable during the validation process and method detection limits obtained by standard deviations of field blanks. To calculate the uncertainty matrix, the formula reported in the PMF 5.0 guide."*

R: Finally, for PMF solution the major Saharan dust event having almost one order higher concentrations than any other datapoint must be removed from the dataset as it is another driving force that distorts the solution.

*A: The Saharan dust is one the most important source and one of the aim of this paper is demonstrated the presence of this events in the high mountain in the Eastern Italian Alps. This is the first time that these events are chemically characterized in this region. The PMF allows to recognize several Saharan Dust events, not only the most obvious event of March, and this is also confirmed by the back-trajectories. Of course, PM deeply load in this factor but also some crustal elements are included. If we have to remove the Saharan dust event of March, we produced a bias and this can support also the attribution of the other Saharan Dust events because the factor contribution trend clearly define the presence of these events. Moreover, we considered that a reconstruction of the entire dataset with a $R^2$ of 0.97 a good result for totally described our data. For this reason, we prefer to maintain the entire dataset.*

Minor comments

Line 58 and many other cases: chemical formulas should be typed properly e.g. NH4+ instead of NH4+

*A: Thanks for the suggestion. Changing in the ACP format, we lost superscripts and subscripts. We tried to correct all manuscript.*

Lines 74-75: preheating of filters seems at edge, ACTRIS recommendation is 3 hours at 800°C, 4h at 400°C may not be enough, but especially for elemental carbon that is not measured here.

*A: We have validated all our analytical methods using filter pre-combusted at 400°C for 4h. The blank values are always evaluated and subtracted to the concentration's values.*

Line 75: At this first occasion it is not clear if 96 h means that each sample was sampled for 96 h or any shorter time every 4th day. Please correct.

*A: We modified the sentence as: "Each sample was collected for 96 h with an average volume of 335 $\pm 12\ m^3$ (at ambient conditions) because this time resolution is demonstrated the best balance between quantifying the target species at trace levels and sampling resolution."*

Lines 125: the sentence: "The elevation of the site was considered the elevation of site (2543 m a.s.l)…" should be corrected

*A: Sorry for the mistake. We modified as: "The elevation setup of starting location was considered as the elevation of site (2543 m a.s.l) plus an extra 1000 m to avoid problems from the surrounding orography."*

Line 133: Concentrations are given at µg/m3. It should be stated clearly whether volumes are real one or standard volumes.

*A: We introduced in the line 75 also the mean sampling volume at ambient conditions. This affirmation clarifies that volume considered is the real volume.*

Line 380 – ssCa is used here as reference element for marine enrichment factors. However, it is not clear at all how ssCa was calculated and if it is even possible when a lot of calcium carbonate is probably present that become soluble in acidic environment on the filter as mentioned above.

*A: As suggested by referee, we introduced a clarification about sea-salt Ca : "ssCa, sea salt derived Ca (calculated as [Ca]-(0.038\*[Na])". This type of calculation is usually applied to the marine enrichment factors (MEFs). It is usually applied to compare our results with literature.*

Line 384-385 – LREE, MREE and HREE are not defined before)

*A: We introduced in the main manuscript the acronyms but in any case Commonly used REE abbreviations are: REE = Rare Earth Elements; RE = Rare Earth; REM = Rare Earth Metals; REO = Rare Earth Oxides; REY = Rare Earth elements and Yttrium; LREE = Light Rare Earth Elements (Sc, La, Ce, Pr, Nd); MREE = Medium Rare Earth Elements (Sm, Eu, Gd, Y, Tb, Dy); HREE = Heavy Rare Earth Elements (Ho, Er, Tm, Yb, Lu); TREO = Total Rare Earth Oxide; TREM = Total Rare Earth Metal.*

Fig 9 – it is difficult to find blue point due to very small data points

Fig 9 and Fig 10 – bigger legend and number font is needed, it is difficult to read it.

*A: As suggested by reviewer, we modified the Figures 9 and 10, now 8 and 9. Moreover, we also modified the figure 11, now figure 10.*

Conclusions should be corrected after major revisions.

*A: We corrected and added some sentences in the conclusion.*

**Anonymous Referee #2**

This study presents a thorough discussion on the chemical composition of PM10 at Col Margherita. The datasets and analysis are robust and provide valuable insights for high-altitude background sites. The source apportionment results are interesting, as mass trajectories from different regions have distinct chemical characteristics. The manuscript can be further improved in the following aspects:

R: The abstract only included the meaning of the study and the method. The main conclusions were not mentioned.

*A: As suggested by reviewer, we improved the abstract with the sources identified by PMF: "Some diagnostic ratios were applied, but source apportionment using Positive Matrix Factorization was used to define the main inputs of PM10 collected at this high-altitude site, resulting in the identification of four factors: 1) Saharan Dust Events; 2) long-range marine/anthropogenic influence; 3) biogenic sources; 4) biomass burning and anthropogenic emissions. It can be inferred that, despite the distant location of the Col Margherita site, both regional and long-range anthropogenic pollution have discernible effects on this area."*

It seems only ~30 compounds were used in the PMF analysis, despite that ~100 is measured. What's the reason? What would happen like if all the measured species are included?

*A: We totally agree with referee, because it was mandatory to explain why we have reduced the number of species. So we have included the follows sentence: "The number of species were reduced, considering that some species have the same source: for example, the sum of L-free amino acids, D-amino acids and photodegradation products of α-pinene (pinic and pinonic acids) are considered for the biogenic sources, the sum of phenolic compounds described the biomass burning input, the sum of carboxylic acids described the secondary organic aerosol, some sugars are included, while only some TE and REE are included in the PMF to avoid to introduce noise in the model. Magnesium and sodium are included as ionic and total species because these elements have different sources."*

Figure 13, the PMF results only show contributions from primary sources and are mostly transported from other regions, are there secondary sources or background aerosols?

*A: We completely agree with the reviewer, because the F3 also included the secondary sources. For this reason, we modified the name of factor, and we introduced one sentence in the main manuscript: "Moreover, the presence of CA suggested that in this factor the contribution of secondary organic aerosol is also included."*

Section 3.2 provides too many details and some may not be relevant to the main conclusions of the manuscript, these parts can go to the supplementary information. For instance, Figure 7-11 provides too many information on the trace elements and it distract the readers a lot.

[revised manuscript text omitted]

The definition of periods are obscure, please define in the method which month are regarded as "spring", "summer", "autumn", "winter", "the seasons without snow cover", "the snow season", "the cold period".

*A: We refer to the astronomical seasons and we added this adjective in the manuscript to clarify the periods. The season without snow cover refers to the specific period without snow.*

Figure 2, what is "CA", why it increases in Spring and Summer?

*A: We included a small sentence "probably due to an improving of solar radiation and then atmospheric photo-reactivity."*

Line 161-162 please give a brief preview here of what might be missing in December and January, and mention the exact section that discuss this in detail. It's hard for the readers to search for the clues in the long discussion part.

*A: As suggested by referee, we removed the general sentence and we specified as follows: "In December and January, the percentages of explained mass were found to be between 4% and 25%, and in the next section a cationic deficit was demonstrated probably due to the presence of hydrogen cations and then an acid atmosphere."*

Line 173, please give a brief ratio of the concentration measured by the two methods.

*A: We added this sentence: "In general, the mean percentage of ionic sodium, magnesium and calcium are 54±45%, 29±27% and 65±45%, respectively, compared to the total element (Fig. S3)."*

Line 266-268 The logic here is hard to understand. Authors say that the two acids are from α-pinene, then they say its poorly understood? And the following examples in Line 268-230 should provide specific conclusions of the two studies.

*A: We tried to clarify the conclusions about these species. We removed this sentence because it was confused: "The quantification of the contribution of its sources is still poorly investigated."*

*We integrated this part with the comparison of Kawamura review and our results as follows: "Kawamura et al. (2016) reviewed the main sources and transformations, suggesting that significance of atmospheric photochemical oxidation processing with a similar temporal trend with solar irradiation and ambient temperatures. It is confirmed by our results where the higher concentrations are found during the spring and summer season, while during winter the lowest concentration are quantified (Fig. S5)."*